# Cu-Doped MnO_2_ Catalysts for Effective Fruit Preservation via Ozone Synergistic Catalytic Oxidation

**DOI:** 10.3390/foods13244127

**Published:** 2024-12-20

**Authors:** Jianguo Huang, Rashid Khan, Chunhui Zhai, Xianting Ding, Li-Sha Zhang, Jin-Ming Wu, Zhizhen Ye

**Affiliations:** 1State Key Laboratory of Silicon and Advanced Semiconductor Materials, School of Materials Science and Engineering, Zhejiang University, Hangzhou 310027, China; 12026093@zju.edu.cn (J.H.); lisha1998525430@163.com (L.-S.Z.); yezz@zju.edu.cn (Z.Y.); 2Zhejiang Provincial Key Laboratory of Advanced Chemical Engineering Manufacture Technology, College of Chemical and Biological Engineering, Zhejiang University, Hangzhou 310027, China; rashid@mail.ustc.edu.cn; 3Department of Anesthesiology and Surgical Intensive Care Unit, Xinhua Hospital, School of Medicine and School of Biomedical Engineering, Shanghai Jiao Tong University, Shanghai 200240, China; chunhuizhai@sjtu.edu.cn (C.Z.); dingxianting@sjtu.edu.cn (X.D.); 4State Key Laboratory of Oncogenes and Related Genes, Institute for Personalized Medicine, Shanghai Jiao Tong University, Shanghai 200240, China

**Keywords:** ozone synergistic catalytic oxidation, fruit preservation, ozone degradation, ethylene, bacteria, viruses

## Abstract

Developing and implementing technologies that can significantly reduce food loss during storage and transport are of paramount importance. Ozone synergistic catalytic oxidation (OSCO) technology has been developed, which sterilizes bacteria and viruses on the surface of food and degrades ethylene released during fruit storage through the active oxygen produced by the catalytic decomposition of ozone. Herein, we report the hydrothermal synthesis of MnO_2_ with distinct phase compositions and nanostructures through simply varying the reaction temperatures. Optimized copper-doped α-MnO_2_ nanorods exhibited remarkable efficacy in activating ozone at a concentration of 40 ppb, and this activation resulted in the complete eradication of indicator bacteria on food surfaces within a 24 h period. Moreover, these nanorods demonstrated high effectiveness in decomposing more than 80% of the ethylene molecules emitted by apples and bananas during the preservation period. The high concentration of surface oxygen vacancies is believed to contribute to the enhanced catalytic activity of the Cu-doped α-MnO_2_ catalyst in the OSCO procedure by reducing ethylene production and maintaining the fruit quality during the preservation period.

## 1. Introduction

Global food security and sustainable environmental development have long been major challenges for humanity [1,2]. There are still 691–783 million hungry people worldwide, and during food production, processing, and storage, bacterial growth combined with ethylene accumulation leads to decay and spoilage, resulting in significant food loss [3,4,5]. Ripening is a complex and irreversible biological process regulated by genetic programming that encompasses various alterations in physiological and biochemical properties. Fruits and vegetables emit ethylene, which is associated to their development, maturation, and sprouting. The specific effects of ethylene depend on factors such as the type of produce, stage of development, and the duration of ethylene exposure. Ethylene not only hastens fruit ripening, but can also cause excessive ripening and deterioration, thereby shortening shelf life and resulting in waste. Furthermore, there is a growing need to preserve food quality for extended periods and minimize spoilage, driven by both health concerns and economic considerations [6]. Therefore, for post-harvest preservation, it is crucial to use inhibitors or scavengers of ethylene synthesis to delay the ripening process in fruits and vegetables.

Traditional food preservation methods include chemical [7], physical [8], and biological [9] methods. Chemical preservation techniques [10,11] mainly include the use of preservatives, coating techniques, and acid-electrolyzed water, which have the advantages of simplicity, speed, and effectiveness. However, environmental harmful substances and residues pose potential threats to human health. Physical preservation techniques include low-temperature preservation, vacuum preservation, radiation preservation, heat treatment, and controlled atmosphere preservation, among which low-temperature storage is widely used; however, there are problems such as frost, high energy consumption, and ineffective treatment of food surface bacteria [12]. Biological preservation includes various routes, including biological antagonistic bacteria, genetic engineering, and enzyme technology. However, the safety of these technologies with respect to human health remains uncertain [13].

Preserving fruits serves several important purposes, such as reducing food waste and allowing people to access the nutritional value of fruits even when fresh fruits are unavailable. This process is crucial for maintaining a consistent supply of fruit-based nutrients for consumers. Furthermore, the process of preservation allows fruits to be available throughout the year, regardless of seasonal fluctuations in fruit production [14]. This is particularly beneficial in regions where certain fruits are not grown locally or only obtainable during specific seasons. Ozone has broad-spectrum antibacterial effects on food surfaces and has been used to preserve many fruits and vegetables [15,16]. However, it produces toxic by-products and excessive inhalation poses risks to the respiratory systems of humans and animals [17,18]. Manganese dioxide (MnO_2_) has been extensively investigated as a potential catalyst to supplant precious metals owing to its superior redox, ion-exchange, and molecular adsorption properties [19]. The diverse oxidation states of manganese (Mn^2+^, Mn^3+^, and Mn^4+^) and their respective geometries potentially contribute to their synergistic capacity to catalyze the oxidation process [20]. Moreover, compounds based on manganese oxide with diverse Mn oxidation states contain numerous oxygen vacancies, which can improve the generation and activation of the oxidized species [21]. The different crystalline phases of MnO_2_, such as β, γ, δ, and α-MnO_2_, result from different configurations of the [MnO_6_] octahedral structure [22]. The differences in their tunnel configurations, oxygen species on the surface, and redox characteristics lead to significant variations in their performances during catalytic oxidation [23]. However, an efficient and stable MnO_2_-based catalyst with a high number of oxygen vacancies is still lacking [24]. In particular, α-MnO_2_ demonstrated superior catalytic efficiency compared to other phases of MnO_2_ [25]. The significant interactions between Mn and other incorporated metals, such as Ce, Cu, Co, Fe, and Ni, can enhance catalytic efficiency by altering the microstructure and facilitating the redox cycle through electron transfer [26,27]. Additionally, the addition of doped metal ions can effectively regulate the concentration of oxygen vacancies in MnO_2_ [27,28].

In response to the limitations of the existing ozone preservation technology, we herein report a novel approach based on the use of Cu-doped MnO_2_ materials and a synergistic catalytic oxidation reaction with ozone. This ozone synergistic catalytic oxidation (OSCO) technology presents a promising solution for industry by enabling the rapid breakdown of ozone into reactive oxygen species. In the current investigation, highly efficient Cu-doped MnO_2_ produces oxygen free radicals in the presence of trace ozone, resulting in breakdown and penetration of the cell wall. This leads to the leakage of small molecules, such as ions, degradation of internal components, and, ultimately, inactivation of bacterial and viral particles. Additionally, this methodology decelerates the ripening process of fruits through ethylene degradation and significantly prolongs their shelf life. Compared with other ozone technologies, this method does not leave ozone residues in the air, thereby improving its efficacy and safety [29]. This technique can be used to disinfect and preserve food in various environments, thereby extending the shelf life of food more than 10 times.

## 2. Experimental Section

### 2.1. Synthesis of Cu/MnO_2_ Catalyst

The Cu-doped MnO_2_ catalysts were prepared by a hydrothermal method, as schematically illustrated in Figure 1a. In a typical synthesis, 400 mL of deionized water was added in a beaker and stirred at 400 rpm, and then 30 g of KMnO_4_ (Aladdin Biochemical Technology Co., Shanghai, China) and 10 g of MnSO_4_∙H_2_O (Aladdin Biochemical Technology Co., Shanghai, China) were introduced and allowed to react for half an hour. Subsequently, 10 g of CuSO_4_∙5H_2_O (Aladdin Biochemical Technology Co., Shanghai, China) was added and the mixture was further stirred under ambient conditions for 1 h. The slurry was then transferred to an autoclave with a temperature maintained at 100, 120, 140, or 160 °C, respectively. After 12 h, the reaction was complete and the mixture was cooled to room temperature, followed by filtration and separation. The resulting product was dried at 150 °C for 10 h to obtain catalyst powder.

### 2.2. Honeycomb-Shaped Catalyst Fabrication

The Cu-doped MnO_2_ catalyst (40 g) was thoroughly mixed with 400 mL water and stirred for 30 min. During stirring, 0.1 g of carboxymethyl starch (CMS) (Aladdin Biochemical Technology Co., Shanghai, China), 5 g of neutral silica sol (30%) (Zhejiang Delixin Micro-nano Technology Co., Ltd., Shanghai, China), and 0.1 g of dispersant (cetyl trimethylammonium bromide (Aladdin Biochemical Technology Co., Shanghai, China)) were added to the slurry. After 30 min of further stirring, the mixture was transferred to a milling bowl in a planetary ball milling machine with a rotation speed of 400 rpm and milled for 60 min. A white cordierite ceramic honeycomb (diameter 60 mm, pore size 1 × 1 mm, 25 g in weight, (Pingxiang OceanPower Chemical Co., Ltd., Pingxiang, China)) was soaked in the prepared slurry for 20 min until no bubbles were generated, and then the residual slurry was removed using an air compressor. After 30 min of natural drying, the soaked honeycomb was placed in an oven (110 °C) for 12 h to obtain the final loaded ceramic honeycomb. The preparation method of the honeycomb catalyst is shown schematically in Figure 1b. The mass loading of the catalyst was 6 mg/g.

### 2.3. Materials Characterizations

The morphologies of the catalyst powders were characterized by scanning electron microscopy (SEM, Quanta 400, FEI) and transmission electron microscopy (TEM). Energy-dispersive spectroscopy (EDS) was performed using a TALOS F200X apparatus from Thermo Fisher, United States (Waltham, MA, USA). The crystal structures of the catalysts were determined by X-ray diffraction (XRD) using an X’Pert PRO (PANalytical, Almelo, The Netherlands) instrument at room temperature with Cu K_α_ radiation (λ = 1.5418 Å). The chemical compositions and valence states of the catalysts were measured by X-ray photoelectron spectroscopy (XPS) using a K-Alpha electron spectrometer (Thermo Scientific, Waltham, MA, USA) with an Al Kα radiation source (hv = 1486.6 eV). The specific surface area was evaluated using the Brunauer–Emmett–Teller (BET) method. The testing conditions involved low-temperature N_2_ adsorption–desorption measurements conducted at 77 K using an Autosorb-1-C apparatus from Quantachrome Instruments, Boynton Beach, FL, USA. The pore size distributions were evaluated using the Barrett–Joyner–Halenda (BJH) method.

### 2.4. Antibacterial Testing

Antibacterial testing was conducted within a testing chamber, employing *Escherichia coli* (*E. coli*) as the indicator bacteria for disinfection tests. The step-wise procedure is provided in the online Appendix A.

### 2.5. Food Preservation and Ethylene Concentration Testing

A detailed food preservation process using Cu-doped MnO_2_ catalysts is schematically shown in Figure 1c. The same food items were placed in two identical chambers, serving as the experimental and control chambers. The temperature in both chambers was set to 25 °C and the relative humidity was maintained at 70%. In the experimental chamber, 275 g of the honeycomb-shaped catalyst with a total volume of 327 cm^3^ was installed; for the control, only ozone was used. The airflow rate was set to 35 m^3^/h. Every 24 h, the ethylene concentration inside the testing chamber was measured remotely using an ethylene detector (NEO, MP-180, Canton, NC, USA). For ethylene monitoring, each chamber contained 5 apples and 10 bananas (see Appendix A and the corresponding section in the Appendix A for further information).

### 2.6. Antimicrobial Efficacy Testing

An ozone generator employing dual-quartz dielectric technology was utilized to produce ozone with a maximum output of approximately 5 g/h. An ozone analyzer (MODEL 202, 2B Technologies Inc., Broomfield, CO, USA) was employed to measure the ozone concentration with an accuracy of ca. 0.01 ppm. The apparatus for assessing antimicrobial efficacy on food surfaces primarily consists of a Programmable Logic Controller (PLC, Smart, Siemens, Munich, Germany), ozone generator (YH-1, Fengsuo, Xuzhou, China), time relay, and air humidifier, as schematically illustrated in Figure 2. The ozone concentration of 40 ppb was regulated by a time relay. Air humidity is controlled by the PLC, which governs the operation of the air humidifier. The test temperature ranged from 1 °C to 30 °C, and the tests were conducted in a biochemical laboratory.

## 3. Results and Discussion

The XRD measurements were performed to analyze the phase structures of the Cu-doped MnO_2_ catalysts synthesized at various hydrothermal temperatures, as show in Figure 3a. The XRD peaks of the four different phases, α, β, δ, and γ-MnO_2_, correspond to the standard PDF cards of JCPDS 44-0141, 24-0735, 80-1098, and 14-0644, respectively. No additional peaks were observed in any of the four crystal phases of Cu-doped MnO_2_, further indicating that copper was well incorporated into the structures. The hydrothermal temperature plays a crucial role in the crystal phases of the catalysts. At temperatures of 100 °C or lower, the γ-MnO_2_ phase was obtained, which was converted to the α-MnO_2_ phase when the temperature was increased to 120 °C. When the temperature was increased to 140 °C, the α-MnO_2_ phase underwent a structural change, transitioning from a (2 × 2) tunnel structure to a (1 × 1) arrangement, which is characteristic of β-MnO_2_. Moreover, β-MnO_2_ transformed into δ-MnO_2_ phase when the temperature was further increased to 160 °C. In a word, the phase structures of MnO_2_ hydrothermally synthesized at temperatures of 100, 120, 140, and 160 °C were γ, α, β, and δ, respectively, as listed in Table 1.

The SEM images of the Cu-doped MnO_2_ catalysts synthesized at different temperatures are depicted in Figure 3b–e. As can be seen in Figure 3b, α-MnO_2_ exhibited a thick nanorod structure, while β-MnO_2_ showed almost the same nanorod structure but was relatively thinner and longer (Figure 3c). δ-MnO_2_ displayed flower-like structures in which the nanosheets were interconnected, as can be seen in Figure 3d. Moreover, the γ-MnO_2_ phase exhibited an urchin-type structure, as shown in Figure 3e. The SEM investigation showed that when the reaction temperature increased, the structure underwent conspicuous transformations, sequentially manifesting morphologies corresponding to α-MnO_2_, β-MnO_2_, δ-MnO_2_, and γ-MnO_2_ [30,31].

The catalytic performance of Cu-doped α-MnO_2_ was found to stand out among the four samples; therefore, it was further characterized by TEM and XPS. The TEM image further indicates a nanorod shape (Figure 4a). The lattice spacing of 0.24 nm corresponds to the (211) plane of the α-MnO_2_ phase, as can be seen in the HRTEM image (Figure 4b). The Cu, Mn, and O elements were homogeneously distributed in the Cu-doped α-MnO_2_ catalyst, as confirmed by EDS elemental mapping (Figure 4c–f).

The chemical states of the Cu-doped α-MnO_2_ catalyst were analyzed by XPS. The survey spectrum contained Cu, Mn, and O, indicating that copper atoms were successfully doped into the MnO_2_ catalyst, as shown in Figure 5a. Figure 5b shows the high-resolution XPS spectrum of Cu 2p, where the two peaks located at binding energies of 934.3 and 954.5 eV corresponded to the Cu 2p_3/2_ and Cu 2p_1/2_ electronic states, respectively, suggesting that copper is primarily in the divalent valence state. Moreover, the divalent state was further confirmed by two satellite peaks observed at 945.2 and 962 eV [32]. The Mn 2p high-resolution XPS spectrum exhibited two major peaks at 642.2 and 653.9 eV associated with the valence states of Mn 2p_3/2_ and Mn 2p_1/2_ (Figure 5c), respectively. The valence state of Mn 2p_3/2_ splits into two peaks positioned at 642.2 (Mn^3+^) and 644 eV (Mn^4+^), implying that the α-MnO_2_ phase contains both Mn^3+^ (0.645 Å) and Mn^4+^ (0.53 Å) ions [33,34]. The contents of Mn^3+^, Mn^4+^, and the molar portion of surface Mn^3+^/Mn^4+^ in Cu-doped α-MnO_2_ are provided in Appendix A. Copper doping could replace Mn^4^⁺ or Mn^3^⁺ ions and increase the number of oxygen vacancies because of the substitution of larger Cu^2^⁺ ions (0.73 Å), causing imbalances in the system. The difference in spin energy between the two peaks was 11.7 eV, which is in good agreement with previously reported values [35]. The O 1s spectrum (Figure 5d) exhibited three peaks at binding energies of 529.6, 531.3, and 532.6 eV, corresponding to lattice oxygen (O_latt_), oxygen defects, particularly oxygen vacancies (O_v_), and surface-adsorbed water, respectively [33,36]. The oxygen vacancy content of the Cu-doped α-MnO_2_ catalyst was 31.62% (Appendix A), which is significantly higher than that of the Cu-doped δ-MnO_2_ catalyst (21.45%). Copper doping in MnO_2_ catalysts enhances the number of oxygen vacancies which are responsible for their high catalytic activity. Moreover, the Mn^3+^ content and higher number of oxygen vacancies in the catalyst also play a vital role in the catalytic performance of the Cu-doped α-MnO_2_ catalyst [37]. In the current investigation, the Cu-doped α-MnO_2_ and δ-MnO_2_ catalysts demonstrated the most significant bactericidal efficacy compared to the β-MnO_2_ and γ-MnO_2_ catalysts. To find the differences in the chemical states of the MnO_2_ catalysts, the Cu-doped δ-MnO_2_ catalyst was also investigated by XPS, as shown in Appendix A. The Cu-doped α-MnO_2_ catalyst exhibited outstanding catalytic activity compared to that of Cu-doped δ-MnO_2_ because of the high Mn^4+^/Mn^3+^ molar portion and the higher number of oxygen vacancies (Appendix A). Moreover, charge imbalance in the system induces more oxygen vacancies and produces electron enrichment [36,38].

The Brunauer–Emmett–Teller (BET) specific surface area and pore structure of the catalyst significantly influence its catalytic activity. Low-temperature N_2_ adsorption–desorption experiments were carried out, as depicted in Figure 6a. The α, δ, and β-MnO_2_ catalysts exhibited a typical type IV isotherm with a broad relative isotherm pressure value between 0.4 and 1.0, implying a mesoporous structure according to IUPAC classification [39]. However, the relative isotherm pressure range became narrower for the γ-MnO_2_ catalyst (0.6–1.0). The pore size distribution of the catalysts was calculated from the Barrett–Joyner–Halenda (BJH) method, as shown in Figure 6b. These findings indicate that the α, δ, and β-MnO_2_ catalysts exhibit a considerably narrower pore-size distribution, falling within the range of less than 4 nm. In contrast, the pore-size distribution of the γ-MnO_2_ catalyst was characterized by discernible differences, which aligns well with the results of the N_2_ adsorption–desorption analysis. The BET specific surface area and mesoporous parameters are displayed in Table 1. δ-MnO_2_ has the largest specific surface area of 97.7 m^2^/g, while α-MnO_2_ has a smaller surface area of 73.6 m^2^/g. The average pore diameter of α-MnO_2_ was approximately 10.6 nm, which was larger than that of δ-MnO_2_ (7.5 nm). The pore volume of α-MnO_2_ is about 0.18 cm^3^ g^−1^, slightly smaller than that of δ-MnO_2_ (0.19 cm^3^ g^−1^). It was found that the hydrothermal temperature plays a crucial role in the crystal phase and morphology and changes the pore size of MnO_2_.

The Cu-doped α-MnO_2_, β-MnO_2_, δ-MnO_2_, and γ-MnO_2_ were utilized along with blank group to test the bactericidal efficiency against *E. coli*, as shown in Figure 7a. It can be observed that Cu-doped α-MnO_2_ and δ-MnO_2_ honeycomb catalysts demonstrated the most significant bactericidal efficacy against *E. coli* as compared to β-MnO_2_ and γ-MnO_2_ catalysts. Interestingly, the germicidal efficiency of the Cu-doped α-MnO_2_ honeycomb catalyst reached 99.9% in only 24 h. The incorporation of copper into MnO_2_ and the crystal phase both play important roles in improving the catalytic activity of the catalyst [31]. XPS analysis provided direct evidence of water adsorption on the surface and the presence of oxygen vacancies in the Cu-doped α-MnO_2_ catalyst. This strongly suggests that Cu-doped α-MnO_2_ efficiently activates adsorbed oxygen, generating active oxygen species, which play crucial roles in deactivating *E. coli* [40]. These results demonstrate that Cu-doped α-MnO_2_ is a potential catalyst owing to its crystal phase and abundant oxygen vacancies on its surface.

The shelf-life extension and rotting prevention of various foods were tested, as shown in Figure 7b. Here, the freshness time refers to the shelf-life of food without causing weight loss or injury, which is determined by observations of specific signs of decay: substantial dark patches forming on banana peels, decomposition occurring at the base of the fruit, and emergence of fungal growth on the surface of oranges. During this period, food remains edible and safe for consumption. The Cu-doped α-MnO_2_ catalyst demonstrated the most excellent bactericidal efficacy against *E. coli* as compared to β-MnO_2_, γ-MnO_2_, and δ-MnO_2_ catalysts. Therefore, a Cu-doped α-MnO_2_ catalyst was used to test the shelf life of different items, such as bananas, oranges, durian shells, leafy vegetables, and pork. To test the shelf-life extension, a test chamber with a size of 1 m^3^, temperature of 25 °C, humidity of 70%, and ozone concentration of 40 ppb was used. Upon interaction with a catalyst, ozone undergoes catalytic decomposition and rapidly produces oxygen radicals. Subsequently, this intermediate collides with the ozone molecules, resulting in the production of oxygen molecules [23]. The catalytic ozone decomposition mechanism is represented by the following reactions [41]:(1)$$O_{3}+V_{o}\;\to \;O_{2}+O^{2-}$$
(2)$$O_{3}+O^{2-}\;\to \;O_{2}+O_{2}^{2-}$$
(3)$$O_{2}^{2-}\;\to \;O_{2}+V_{o}$$
where V_O_ represents the oxygen vacancies. Upon interaction with ozone, the Cu-doped α-MnO_2_ catalyst facilitates the binding of ozone molecules to the MnO_2_ surface. This process occurs because one of the oxygen atoms from the ozone molecule occupies an available oxygen vacancy site on the catalyst surface. When an oxygen vacancy donates electrons to an ozone molecule, it results in the formation of surface-bound oxygen species (O^2−^) at the vacancy site. Simultaneously, a dioxygen molecule is created, which escapes into the surrounding gas phase. Following this, an additional ozone molecule interacts with O^2−^ binding to the surface, resulting in the formation of O_2_^2−^ and dioxygen molecules. Ultimately, the peroxide compound breaks down and releases oxygen molecules, restoring the oxygen vacancies. This vacancy then takes part in subsequent cycles to further break down the ozone molecules. When peroxide fails to break down promptly, oxygen vacancies cannot be restored, and peroxide is converted to lattice oxygen. Moreover, the incorporation of Cu enhances the oxygen vacancies of the catalyst, which serve as active sites, thus facilitating the overall process of ozone conversion. Furthermore, the incorporation of copper ions into the structure may enhance the regeneration capacity of the oxygen vacancies. This was due to the weakening of the Mn-O bonds in the octahedral MnO_6_ framework, which occurred as a result of the Jahn–Teller effect [33,42]. Consequently, this process can improve the overall regeneration performance. This synergistic process results in a rapid decrease in the ozone concentration to a safe level, leaving no residue in the air. At the same time, the highly reactive intermediate radicals neutralize bacteria and microorganisms while rapidly decomposing ethylene released during storage, thereby prolonging the shelf life of food without causing any weight loss or injury symptoms. Additionally, freshness indicates that the food remains edible and safe for consumption throughout this time period. The OSCO technology can significantly extend the shelf life of various foods by effectively controlling microbial growth and reducing the oxidation processes that lead to food spoilage, thus providing an effective, safe, and energy-efficient method for food preservation.

Ethylene production and fruit softening are associated with the degradation of cell walls, which is facilitated by polygalacturonase and pectin methylesterase. Ethylene production stimulates these enzymes. The action of cell-wall-degrading enzymes is impeded by ozone, which slows the process of pectin dissolution. This, in turn, leads to a reduction in the activity of polygalacturonase and galactosidase. Furthermore, this process reduces pectin methylesterase activity and decreases the rate of depolymerization, leading to a slower breakdown of the cell wall [43]. A testing chamber (refer to Appendix A) containing ozone and Cu-doped α-MnO_2_ honeycomb catalyst was utilized to evaluate the degradation efficiency of ethylene released from 5 apples and 10 bananas, as well as the preservation duration of the food items. In addition, different fruits were used to assess the change in the preservation state, such as oranges, durian shells, peaches, and durian fleshy pulps (Figure 8), demonstrating that the OSCO technology markedly extended the shelf life of the fruits (Figure 8a–d) when compared to the control groups, which contained only ozone (Figure 8e–h). This technology can preserve fruits and vegetables without compromising their microstructural integrity and texture over an extended period of time through the decomposition of ozone, resulting in no ozone residue in the atmosphere. Moreover, a set of oranges with one moldy orange was introduced into the testing chamber to check the rate of mold growth on fresh oranges, as shown in Figure 9. As observed, mold growth increased and propagated to adjacent oranges in the control group (Figure 9a–d), whereas mold growth decelerated and did not extend to adjacent oranges in the presence of the Cu-doped α-MnO_2_ catalyst (Figure 9e–h). The results indicated that OSCO technology successfully extended the shelf-life of the fruits owing to its catalytic performance in conjunction with ozone in degrading ethylene [44]. Ozone-catalytic oxidation technology has been shown to effectively degrade the ripening agent ethylene and simultaneously annihilate airborne pathogens, thereby retarding the ripening and decay of fruits and vegetables [19,45,46]. Consequently, this technology significantly improves the mechanical properties and prolongs the duration of preservation of multiple types of food.

Ozone-assisted catalytic oxidation is a crucial step in superoxide catalysis. It catalyzes the decomposition of ozone generated by ozone generators into active oxygen on the catalyst surface. Ozone generates oxygen-derived free radicals on the surface of catalyst materials, which subsequently react with ethylene, resulting in its degradation [47,48]. On the other hand, active oxygen damages and penetrates the cell walls, causing leakage of ions and other small molecules within the cell, degradation of internal components, and eventual cell mineralization. This results in the inactivation of bacterial and viral particles, thereby delaying the decay process and significantly prolonging the preservation period of fruits and vegetables.

As illustrated in Appendix A in the online SI, apples and bananas were used to perform ethylene degradation experiments. Cu-doped MnO_2_ catalysts exhibiting different crystal phases were utilized for the degradation of ethylene while maintaining an ozone concentration of 40 ppb. The ethylene concentration in each chamber was quantified daily using ethylene detector equipment for up to ten days. The optimized copper-doped α-MnO_2_ nanorods demonstrated high effectiveness in decomposing more than 80% of the ethylene molecules emitted by apples and bananas during the preservation period, as shown in Figure 10. The bananas and apples treated with OSCO technology were fresher than those not treated with OSCO (Appendix A). The ripening phase is characterized by the appearance of dark and tan markings that result from enzymatic browning. This phenomenon occurs when phenolic substances exit the cell and undergo oxidation by the enzyme polyphenol oxidase (PPO) [49]. The pigment responsible for the dark and brown coloration was formed when the quinone compound was polymerized. Additionally, OSCO treatment reduces the expression of certain proteins involved in the production of ethylene. Consequently, the decreased ethylene production rate decelerates pathogen growth and slows the ripening process. This enhanced performance is attributed to two key factors: a high ratio of Mn^4+^ to Mn^3+^ and an increased concentration of oxygen vacancies. Moreover, the Cu-doped δ-MnO_2_ catalyst also performed well in ethylene degradation compared with the other crystal phases, suggesting that the δ-MnO_2_ crystal phase also possesses a high number of oxygen vacancies. In the fruit industry, ethylene is a major factor that accelerates the ripening process, leading to concerns regarding reduced shelf life and over-ripening. This technology effectively reduces ethylene production without affecting the nutrients inside the fruits during the preservation period, ultimately extending shelf life of the fruits. Compared to other ozone-based techniques, there is no ozone residue in the air, which enhances the safety and efficiency of OSCO technology. Moreover, this technology is applicable in various environments for food preservation, improving fruit quality during storage and transportation, and minimizing food waste. Consequently, this offers advantages to both producers and consumers throughout the supply chain.

## 4. Conclusions

In summary, we successfully synthesized Cu-doped MnO_2_ catalysts via an easily scalable hydrothermal method. The catalysts were prepared at different temperatures to tune their crystal structures and control the oxygen vacancies. The Cu-doped α-MnO_2_ phase contained the highest number of oxygen vacancies, as confirmed by XPS analysis, thus exhibiting high disinfection and ethylene degradation efficiency and significantly prolonging the shelf life of various foods. Under 25 °C and 70% humidity, this technology achieved a 99.9% disinfection efficiency against *Escherichia coli* on the food surface and over 80% ethylene degradation efficiency within the fruit storage space using an ozone concentration of only 40 ppb. Compared to existing food-preservation technologies, this approach exhibits higher disinfection activity and can achieve secure and sustained disinfection. This ozone-assisted catalytic oxidation technology offers a safe and reliable method for food preservation with significant implications for practical applications.

## Figures and Tables

**Figure 1 foods-13-04127-f001:**
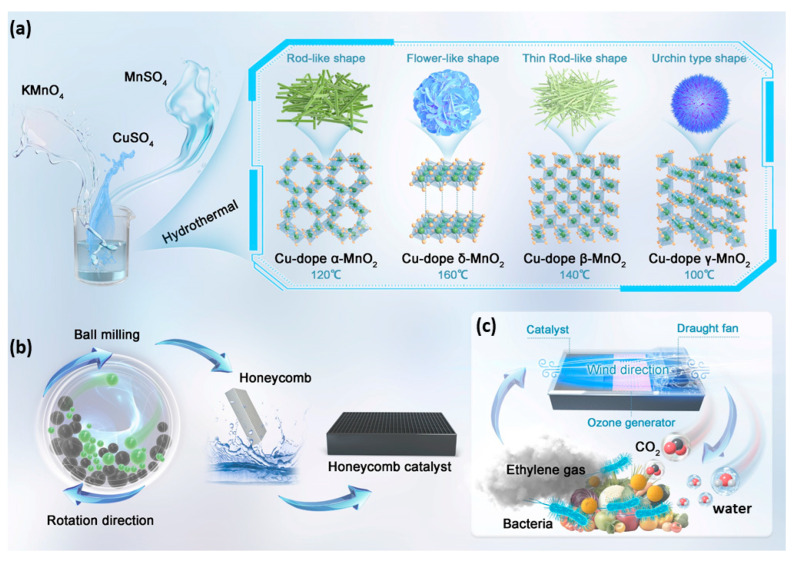
Schematic description of the synthesis of (**a**) Cu-doped MnO_2_, (**b**) honeycomb catalyst, and (**c**) food preservation process.

**Figure 2 foods-13-04127-f002:**
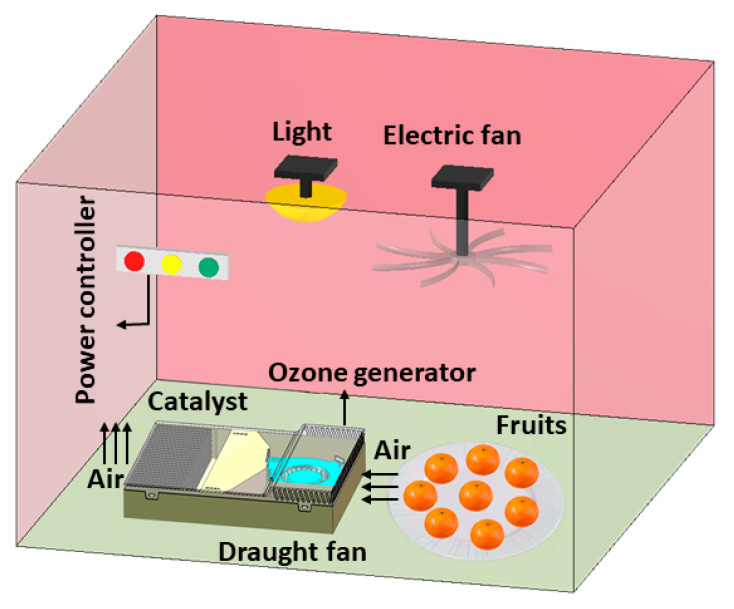
Schematic of the OSCO performance-testing chamber.

**Figure 3 foods-13-04127-f003:**
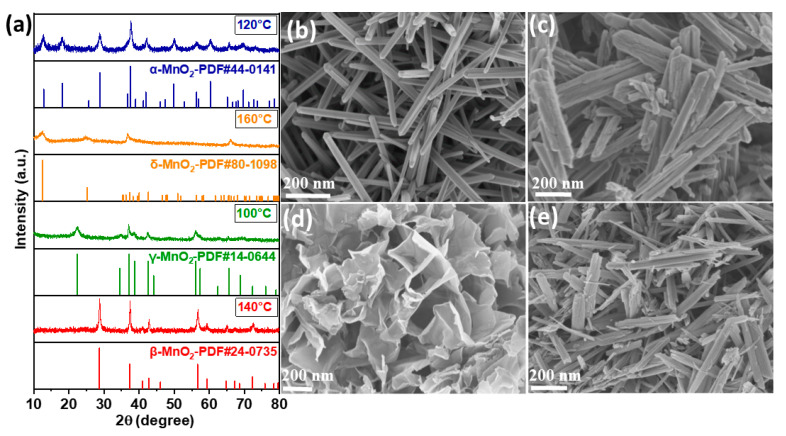
(**a**) XRD patterns of Cu-doped α, δ, γ, and β-MnO_2_ catalysts derived at various hydrothermal temperatures, together with the corresponding JCPDS cards; SEM images of Cu-doped (**b**) α-MnO_2_, (**c**) β-MnO_2_, (**d**) δ-MnO_2_, and (**e**) γ-MnO_2_ catalysts.

**Figure 4 foods-13-04127-f004:**
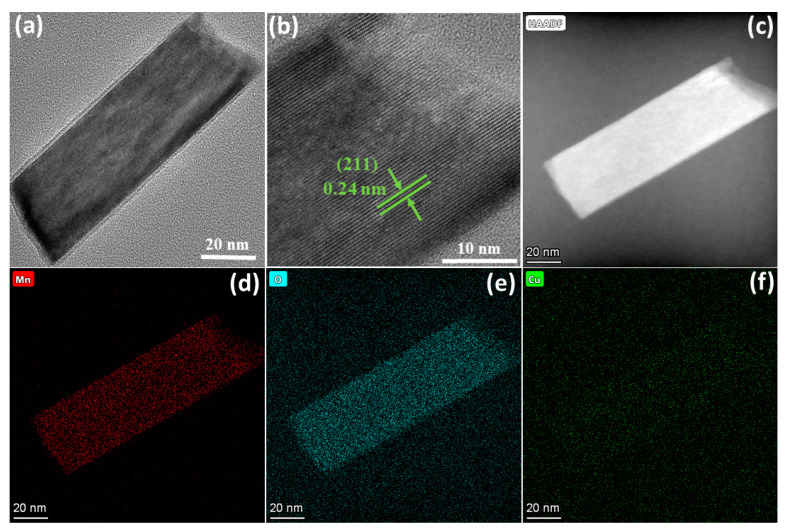
(**a**) TEM, (**b**) HRTEM image of Cu-doped α-MnO_2_; (**c**) HADDF image, elemental distribution of (**d**) Mn, (**e**) O, and (**f**) Cu atoms.

**Figure 5 foods-13-04127-f005:**
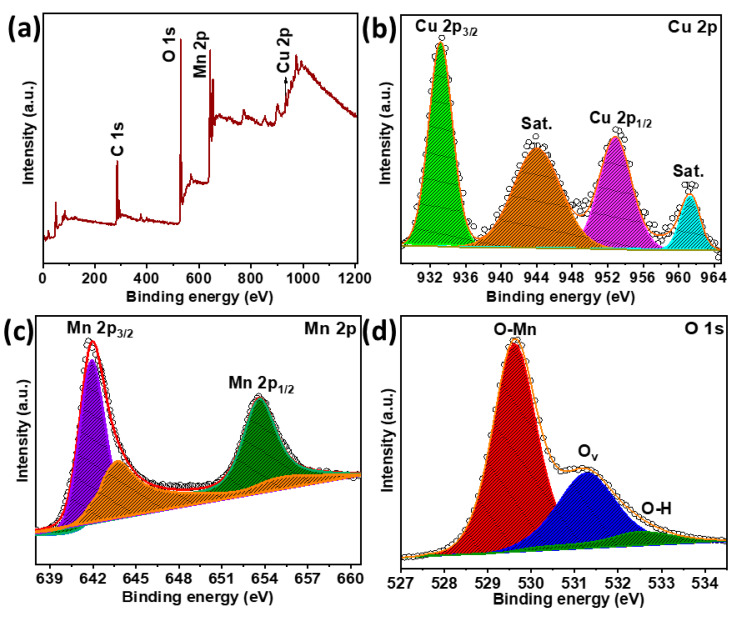
XPS analysis of Cu-doped α-MnO_2_ catalyst: (**a**) survey; (**b**) high-resolution spectra of (**b**) Cu 2p, (**c**) Mn 2p, and (**d**) O 1s.

**Figure 6 foods-13-04127-f006:**
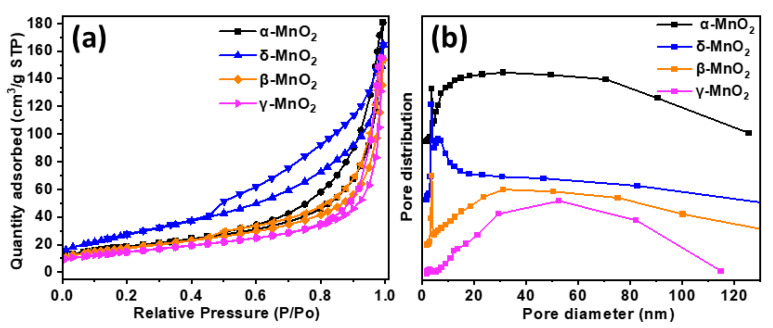
(**a**) N_2_ adsorption–desorption isotherms, (**b**) pore size distribution of Cu-doped α, δ, β, and γ-MnO_2_ catalysts.

**Figure 7 foods-13-04127-f007:**
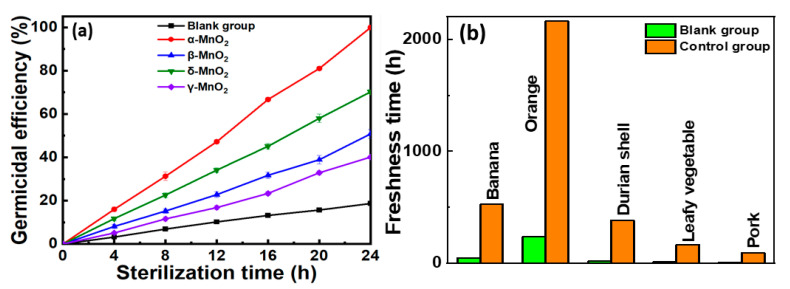
(**a**) Efficacy curves illustrating the antimicrobial performance on food (oranges) surfaces within 24 h activated by Cu-doped MnO_2_ catalysts exhibiting different crystal phases. (**b**) Statistical chart of shelf-life extension of various foods using Cu-doped α-MnO_2_ catalyst in the presence of 40 ppb ozone.

**Figure 8 foods-13-04127-f008:**
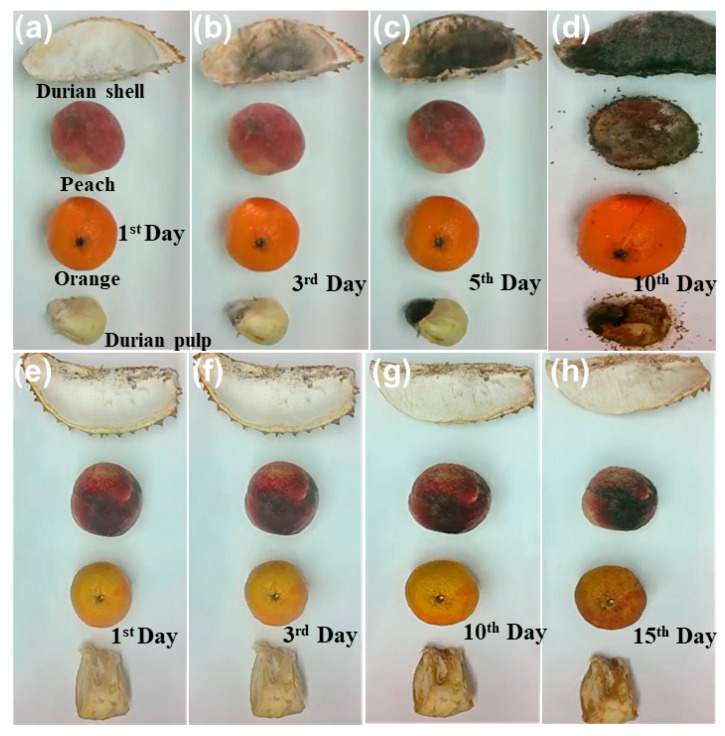
Preservation process of different fruits: (**a**–**d**) control group without OSCO technology and (**e**–**h**) OSCO technology group using Cu-doped α-MnO_2_ catalyst in the presence of 40 ppb ozone. The humidity was 70%, and the temperature was 25 °C. The durations are labelled in the figure.

**Figure 9 foods-13-04127-f009:**
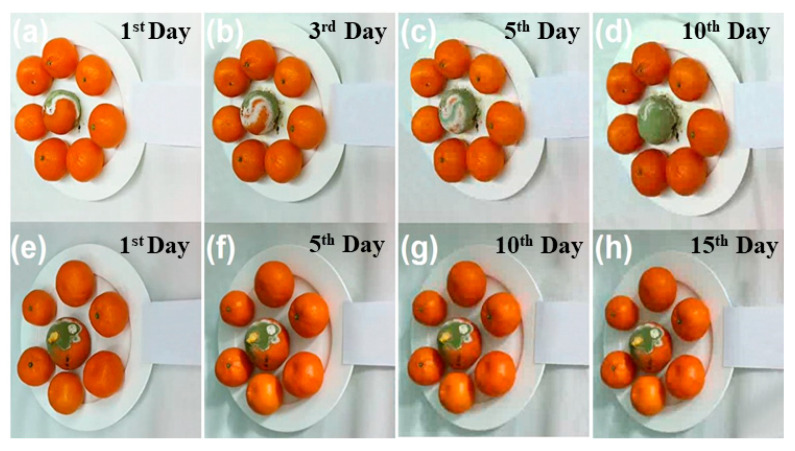
Preservation process of oranges: (**a**–**d**) control group without OSCO technology and (**e**–**h**) OSCO technology group using a Cu-doped α-MnO_2_ catalyst in the presence of 40 ppb ozone. The humidity was 70%, and the temperature was set at 25 °C. The durations are labelled in the figure.

**Figure 10 foods-13-04127-f010:**
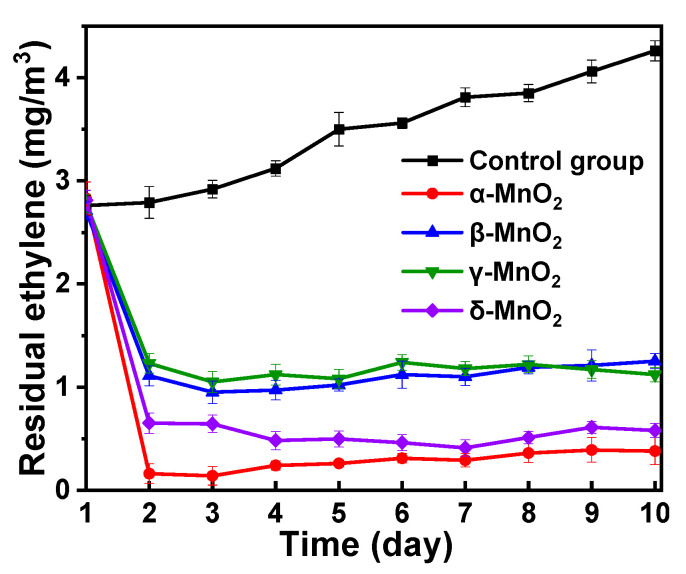
Degradation of ethylene released from apples and bananas using different crystal phases of Cu-doped MnO_2_ catalysts. The error bars were obtained by three repetitive tests.

**Table 1 foods-13-04127-t001:** BET specific surface area, pore size, and pore volume of Cu-doped α, δ, β, and γ-MnO_2_ catalysts with distinct nanostructures synthesized at different temperatures.

Catalysts	Hydrothermal Temperature (°C)	Nanostructure	Specific Surface Area (m^2^ g^−1^)	Pore Volume (cm^3^ g^−1^)	Average Pore Diameter (nm)
α-MnO_2_	120	Coarse nanorods	73.6	0.180	10.7
δ-MnO_2_	160	Nanosheets	97.7	0.190	7.6
β-MnO_2_	140	Fine nanorods	64.2	0.150	9.4
γ-MnO_2_	100	Urchin	53.8	0.128	9.5

## Data Availability

The original contributions presented in the study are included in the article/Appendix A, further inquiries can be directed to the corresponding author.

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
