# Peer review of "Cu-Doped MnO2 Catalysts for Effective Fruit Preservation via Ozone Synergistic Catalytic Oxidation"

_foods, 2024, doi:10.3390/foods13244127_

Round 1

Reviewer 1 Report

Comments and Suggestions for Authors

The work on the Cu-doped MnOâ‚‚ catalyst system to extend the shelf life of food products is of great interest. However, I would like to make a few suggestions to the authors. The methodology for creating the system and evaluating different temperatures for preparing the catalyst to optimise crystalline structures and control oxygen vacancies is comprehensive. However, I think the section on the effects of the catalyst on fruit could be improved. Some statements lack clarity or are not supported by the data.

Line 244-245 states, 'Statistics show that the OSCO technology can significantly extend the shelf life of various foods'. However, it is not clear which statistics are being referenced, nor what evaluation was made. Statistical methods were not mentioned in the methodology.

It is not clear what Fig. 7b refers to. Could you please clarify what is meant by "freshness time"? Could you please clarify how it is measured?

Lines 264-265: 'The results indicate that the OSCO technology successfully extended the shelf-life of the fruits owing to its catalytic performance in conjunction with ozone in degrading ethylene.' However, the study only measured the bactericidal activity (not even clearly) not the shelf-life extension. No assessment was made of the actual extension of shelf life. For instance, weight loss, colour, acid and sugar content, consistency, etc. were not assessed. There was no mention of the impact of the catalyst on fruit quality. The images only demonstrate a reduction in microbial growth, which is insufficient. Please confirm whether the fruit was saleable after 15 days. Please advise how many days the actual shelf life is extended.

Line 281-282: 'The oxygen present in organic compounds (such as ethylene)....' It should be noted that ethylene does not contain oxygen. 

It has been established that there has been a reduction in ethylene accumulation over time. However, it is not clear on which samples the measurement was made. Please confirm whether this applies to all fruits. It should be noted that not all fruits considered are climacteric, and therefore do not produce an increase in ethylene over time. Could you please clarify which item in Fig. 10 is being referenced?

Please clarify whether the degradation of ethylene that you believe you have identified results in a reduction in ripening. Please provide details of the measurement method used. The photos are insufficient for our purposes.

To conclude, there is a discrepancy between the clarity of the system's design and the clarity with which its effectiveness and convenience are presented. There is no advantage in using this system over established refrigeration techniques.

In the case of fruit, shelf life is not simply a matter of reducing microbiological growth, which is usually not a problem. The objective is to preserve internal characteristics, which is best achieved at low temperatures.

Reviewer 2 Report

Comments and Suggestions for Authors

The manuscript Foods-3310375 presents the synthesis, characterization of Cu-doped MnO2 nanocomposites and their application for fruits preservation via ozone synergistic catalytic oxidation. The manuscript is clearly presented, the subject of interest and well supported by various techniques of characterization. Unfortunately, the authors do not indicate what the synergism consists of for the processes that take place. Also, the catalytic processes that take place are not presented more clearly. I therefore recommend the authors to discuss the catalytic processes that take place and the action of nanocomposite materials on the disinfection processes.

Round 2

Reviewer 1 Report

Comments and Suggestions for Authors

It is noticeable that the work you submit for publication in Foods has improved considerably.  I still think it is a very important and definitely interesting paper. However, the part concerning the validation of the OSCO technology on fruit is still very poor, despite the improvements. Without statistical evaluations, it is difficult to show the improvements due to the treatment. The photos are very explanatory, but not enough.

The values presented in Table S2 don't explain anything in a scientific sense. Are the different values recorded different in a statistical sense? TSS, weight, colour, on how many fruits are they made? have repetitions been made? There is no information on texture or firmness (the text states that there is no change in fruit subjected to OSCO). In general, weight is given as weight loss (%) due to transpiration. How do you explain the lower weight loss achieved with your system? with transpiration, ethylene has only a marginal part to do with it. What explanation can you give?  TSS alone is not an important ripening index. The values should be evaluated in conjunction with other parameters (firmness, acidity, color...). The ripening process is a very complex metabolism and to support your conclusion it needs more depth.  I suggest you remove this part and focus on ethylene reduction.
